# CaDrift: A Time-dependent Causal Generator of Drifting Data Streams

## Abstract

This work presents *Causal Drift Generator* (CaDrift), a time-dependent synthetic data generator framework based on Structural Causal Models (SCMs). The framework produces a virtually infinite combination of data streams with controlled shift events and time-dependent data, making it a tool to evaluate methods under evolving data. CaDrift synthesizes various distributional and covariate shifts by drifting mapping functions of the SCM, which change underlying cause-and-effect relationships between features and the target. In addition, CaDrift models occasional perturbations by leveraging interventions in causal modeling. Experimental results show that, after distributional shift events, the accuracy of classifiers tends to drop, followed by a gradual retrieval, confirming the generator's effectiveness in simulating shifts. The framework has been made available on GitHub[1].

## 1 Introduction

In the current era of data, mining high-speed data streams is more important than ever, mainly due to the advent of social media (Yogi et al., 2024), Internet of Things (IoT) (Houssein et al., 2024), and other continuous data sources. Unlike batch-based Machine Learning (ML) configurations, data streams arrive sequentially, posing a possibly infinite flow of data. These streams are often non-independent and identically distributed (iid) and non-stationary, meaning that the data distribution potentially changes over time, a phenomenon known as *concept drift* (or *concept shift*) (Lu et al., 2019). This scenario can be found in a vast variety of research areas, such as healthcare (Jothi et al., 2015), census analysis (Chakrabarty & Biswas, 2018), and fraud detection (Hernandez Aros et al., 2024). Under such circumstances, ML models are expected to perform well on incoming instances, recognize the distributional changes, and adapt accordingly.

Evaluating models under these conditions requires benchmarks that capture not only the non-stationary nature of data but also realistic relationship between features and labels. However, most existing synthetic generators for data streams fall short: they rely mainly on linear or probabilistic functions, and the generated samples are inherently iid, despite concept shift events (Gama et al., 2004; Bifet et al., 2009a; Komorniczak, 2025).

To address these limitations, this work proposes *Causal Drift Generator* (CaDrift), a novel time-dependent synthetic data stream generator with controlled drift events. CaDrift leverages Structural Causal Models (SCMs), commonly used for the generation of synthetic tabular data (Hollmann et al., 2025). To induce time dependence, CaDrift combines exponentially weighted moving average (EWMA) and autoregressive noise on cause-and-effect functions of the causal model, inducing serial correlation across instances. CaDrift can generate an infinite variety of time-dependent tabular datasets with many types of *concept shift* events, including distributional, covariate, severe, and local shifts, in varying rates of change, e.g., abrupt and gradual. Experimental evaluation shows that the proposed framework generates challenging data streams with serial dependence, requiring learners to adapt to new data distributions introduced by changes in causal relationships.

To the best of our knowledge, this is the first work to generate time-dependent synthetic datasets with controlled events of *concept shift* using SCMs. Therefore, the main highlights of this paper are: 1) We present a time-dependent Structural Causal Model (SCM) generator framework capable of generating a virtually infinite combination of synthetic tabular datasets that evolve over time; 2)

---

[1]Available on supplementary material during revision.

CaDrift enables the generation of synthetic data streams with controllable shift events (distributional, covariate, abrupt, etc.) that affect the performance of classifiers.

## 2 BACKGROUND

**Classification.** Classification is a supervised learning task that aims to assign a label $y \in \mathcal{Y} = \{y_1, y_2, \ldots, y_k\}$ to an input $X \in \mathcal{X} \subseteq \mathbb{R}^d$. To do so, a classifier must maximize the *a posteriori* probability $P(y|X)$ of the correct label.

**Concept Drift.** *Concept drift* (Gama et al., 2014) is a prevalent phenomenon in data stream mining. A data stream can be defined as a sequence of instances $S = \{I_1, I_2, \ldots, I_t\}$ such that $I_t = (X_t, y_t)$ corresponds to an instance arriving at time $t$. *Concept drift* occurs when the arriving data distribution changes over time.

*Concept drift* is usually divided into two types (Gama et al., 2014): *real concept drift*, also known as *distributional shift*, and *virtual concept drift*, or *covariate shift*. Other types of drifts are usually derived from these two. Assuming a starting data distribution $P_t(y|X)$ at time $t$, a *distributional shift* happens when $P_t(y|X) \neq P_{t+\delta}(y|X)$ for any $\delta > 0$ (Lu et al., 2019). This means that the probability of a label $y$ being assigned to an input vector $X$ changes over time. Hence, a model trained on the concept at time $t$ may not classify instances on the concept $t + \delta$ properly, bringing the need for models to adapt to *distributional shifts* in a timely fashion.

*Covariate Shift* happens when $P_t(X) \neq P_{t+\delta}(X)$, i.e., the data distribution changes in the feature space but the posterior probability $P(y|X)$ remains unaffected (Lu et al., 2019). As an example, think of an object detection model that has been trained to detect cars. If this model were trained using data only from sunny days, it would never see cars in rainy or snowy conditions. However, the "true" concept definition of what a car is remains unchanged regardless of weather conditions.

*Concept drift* is also categorized depending on the rate of change, where we have abrupt, gradual, and incremental drifts (Lu et al., 2019). *Abrupt concept drift* happens when a change occurs suddenly, in a single time step. Under a *gradual concept drift*, there is a period of coexistence between concepts in which two different distributions arrive in the data stream before the new concept entirely takes place. Lastly, *incremental concept drift* is characterized by a slight change at every time step.

Furthermore, *concept drift* may have a cyclic behavior, often called *recurrent concept drift*, which happens when an old concept returns. The most intuitive example is the change of seasons. Every year, seasonal changes at specific periods (spring, summer, fall, and winter) can be seen as recurrent. For more details regarding *concept drift*, refer to (Bayram et al., 2022; Lu et al., 2019).

## 3 RELATED WORK

**Synthetic Data Generation.** Synthetic data generation is a key tool for evaluating models in controlled environments, as it avoids data privacy concerns while enabling insights into known scenarios. Recently, synthetic generation has gained much attention due to the growing interest in developing Large Tabular Models (LTMs), where access to large and diverse training data is crucial.

A common strategy relies on SCM-based synthetic generators, which model cause-and-effect between nodes. Examples include TabPFN (Hollmann et al., 2025), TabICL (Qu et al., 2025), and Mitra (Zhang & Robinson, 2025). Drift-resilient TabPFN (Helli et al., 2024) is the drift-aware variant of TabPFN, which induces distributional shifts through a second SCM, achieved by modifying edges between nodes. However, this approach still does not explicitly account for temporal dependencies or serial correlation.

In contrast, TabForestPFN (den Breejen et al., 2025), instead of using SCM-based synthetic generators, generates data using tree-based models overfitted on randomly generated features and targets, aiming to expose the model to a wide variety of decision frontiers during training.

Large Language Models (LLMs) have also been utilized for generating tabular synthetic data. Borisov et al. (2023) has presented Generation of Realistic Tabular data (GReaT), an LLM that has been fine-tuned on tabular data and then used to sample synthetic data. Goyal & Mahmoud (2025)

also utilizes fine-tuned LLMs for generating synthetic data from a source dataset, aiming to preserve data privacy. However, LLMs may not be a good approach to generate synthetic tabular data due to tokenization, which implies that each continuous feature is a set of tokens, e.g., "1" $\rightarrow$ "." $\rightarrow$ "15". For that reason, LLMs have been observed not to deal well with continuous features (van Breugel & van der Schaar, 2024). Furthermore, since LLMs are trained on a multitude of popular benchmarks, their reliability in generating synthetic data should be questioned, primarily due to data leakage.

**Concept Drift Generators.** Studies of *concept drift* usually rely on a set of generators, such as SEAConcepts (Street & Kim, 2001), STAGGER (Schlimmer & Granger, 1986), and RandomRBF (Bifet et al., 2009b). These synthetic generators are still widely used today to evaluate classification models under the *concept drift* perspective (Barboza et al., 2025; Guo et al., 2025). More recently, Open World Data Stream Generator with Concept Non-stationarity (OWDSG) (Komorniczak, 2025) introduced *concept drift* on the *Madelon* generator (Guyon et al., 2003) by changing the clusters that define classes.

RealDriftGenerator (Lin et al., 2024) generates synthetic data streams from a source dataset. *Concept drift* is induced through *Clip Swap*, a method that splits the feature values of source datasets into fragments and swaps their positions in the stream. An EWMA is utilized to make the transition between clipped values smoother, thus introducing a drift width.

In most generators, feature values are sampled randomly and do not account for time dependence, such as SEA (Street & Kim, 2001) and Sine (Gama et al., 2004). They can help evaluate how learners react to changes in the decision rule, but do not offer much diversity or complexity.

Thus, there remains a lack of synthetic generators for drifting data streams that can capture complex, high-order relationships and simulate a wide variety of shifts to which learners must adapt. Even though RealDriftGenerator (Lin et al., 2024) claims to simulate synthetic time-dependent drifting data streams, it still needs a source data stream and does not account for causal relationships between features. In contrast, other generators account for random sampling of data instances, and are inherently iid. CaDrift fills these gaps by providing a causal, time-dependent, and synthetic generation of data samples, with controlled drift events that affect causal relationships across features and the target. To the best of our knowledge, this is the first synthetic generator based on SCMs with temporal dynamics across generated samples. In Table 1, we show how CaDrift contrasts with other synthetic generators.

Table 1: Concept matrix of state-of-the-art synthetic generators.

| Method | Causal | Time-dependent | Generates drift | No source |
|---|---|---|---|---|
| STAGGER (Schlimmer & Granger, 1986) | | | X | X |
| SEA (Street & Kim, 2001) | | | X | X |
| Sine (Gama et al., 2004) | | | X | X |
| RandomRBF (Bifet et al., 2009b) | | | X | X |
| TabPFN (Hollmann et al., 2025) | X | | | X |
| Drift-resilient TabPFN (Helli et al., 2024) | X | | X | X |
| TabICL (Qu et al., 2025) | X | | | X |
| TabForestPFN (den Breejen et al., 2025) | | | | X |
| GReaT (Borisov et al., 2023) | | | | |
| (Goyal & Mahmoud, 2025) | | | | |
| RealDriftGenerator (Lin et al., 2024) | | X | X | |
| OWDSG (Komorniczak, 2025) | | | X | X |
| CaDrift | X | X | X | X |

# 4 TIME-DEPENDENT STRUCTURAL CAUSAL MODELS

We propose *Causal Drift Generator* (CaDrift), an SCM-based framework to generate synthetic time-dependent data streams that model high-order relationships between features and targets. Structural Causal Models (SCMs) (Pearl, 2010) models cause-and-effect relationships on Directed Acyclic Graphs (DAGs). First, let us define SCM, as described by Peters et al. (2017):

**Definition 1** *(SCM) An SCM $\mathcal{M}$ with graph $C \rightarrow E$ consists of two assignments:*

$$C := N_c \tag{1}$$
$$E := f_E(C) + N_E, \tag{2}$$

where $N_E$ and $N_c$ are noise variables such that $N_E \perp\!\!\!\perp N_C$. In this model, $C$ stands for cause and $E$ represents the effect. With SCMs, we generate complex cause-and-effect high-order relationships between variables and the target, guided by deterministic effect mapping functions $f_E$ with added Gaussian noise. The advantage of using causal models, as opposed to linear or purely probabilistic models commonly used in existing data stream generators (Gama et al., 2004; Komorniczak, 2025), is their ability to provide information about the consequences of actions (causes) (Pearl, 1995).

It is important to note that, when generating synthetic data streams, one must account for the non-iid nature of data found in a stream. However, the definition of SCM presented considers iid data samples. For that reason, we introduce two components to induce time dependence: 1) an exponentially weighted moving average (EWMA) (Roberts, 1959) to the root nodes distribution, and 2) an autoregressive noise $N_E^{(t)}$ to root and inner nodes. The autoregressive noise induces serial correlation between feature values, while EWMA acts as a smoothing factor for the values propagated in the stream. The EWMA is defined as:

$$Z_t = (1 - \alpha)Z_{t-1} + \alpha X_t \tag{3}$$

such that $Z_t$ denotes the current average, $\alpha \in [0, 1]$ is the smoothing parameter, and $X_t$ the current observation at time $t$. In our framework, the value of $X_t$ is defined by either a Normal or Uniform distribution, since the EWMA is used upon root nodes. Now, in order to introduce the autoregressive noise $N_E^{(t)}$ on the SCM definition, we have:

**Definition 2** *(Time-dependent SCM) The effect function of a time-dependent SCM $\mathcal{M}$ with graph $C \to E$ consists of:*

$$E^{(t)} := f_E(C) + N_E^{(t)}, \tag{4}$$

$$with \quad N_E^{(t)} = \rho N_E^{(t-1)} + \epsilon^{(t)}, \tag{5}$$

where $\epsilon^{(t)} \sim \mathcal{N}(0, \sigma^2)$ is a Gaussian noise, and $\rho \in [0, 1]$ controls the temporal smoothness of the autoregressive noise $N_E^{(t)}$ that makes the next instance in the stream subtly depend on the previous. This allows the noise term to carry memory of past values, introducing temporal correlation across consecutive samples, making generated samples non-iid. The autoregressive noise is applied to all continuous-valued nodes in CaDrift. Given that values of root nodes are assigned through EWMA, by merging Equations 3 and 4, the effect function $f_{x_r}$ of a root node $x_r^{(t)}$ at time $t$, is computed as:

$$x_r^{(t)} = (1 - \alpha)x_r^{(t-1)} + \alpha\theta + N_{x_r}^{(t)} \tag{6}$$

where $\theta$ is sampled from a Normal or Uniform distribution.

**Causal Drift Generator.** In CaDrift, each feature refers to a node in the graph, where each one carries its own mapping function $f_E(C)$, e.g., a small neural network, which defines how parent nodes (causes) influence its value. Figure 1a shows the structure of an SCM node, as represented by a Directed Acyclic Graph (DAG), and Figure 1b depicts an example of a DAG with five features.

To generate a single data sample, the parents' information is passed down to their descendants, where the values of each node depend on the cause-and-effect relationships. Another possibility in the proposed model is to have features depending on the target node. This models relationships where the interest variable can affect other features, such as a disease increasing the number of antibodies and causing symptoms, as also made in previous SCM generators (Hollmann et al., 2025).

The root nodes of the sample DAG ($x_1$ and $x_2$) are initialized by either a Normal or Uniform distribution. The values of inner nodes (e.g., $x_3$ and $x_4$) are defined through a mapping function such as a small neural network. For detailed information about the mapping functions used in this work, see Appendix A.

CaDrift requires the initialization of the mappers before starting the generation of data samples. In contrast to the causal generator presented by Hollmann et al. (2025), which randomly initializes ML models, e.g., MLP, decision tree; our generator, in addition to the random initialization of small

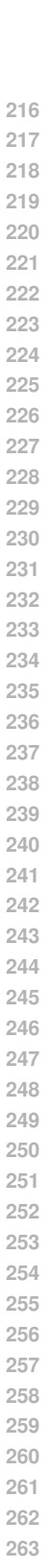
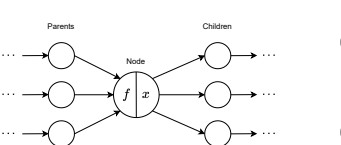
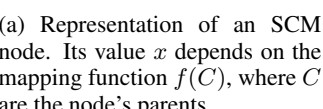
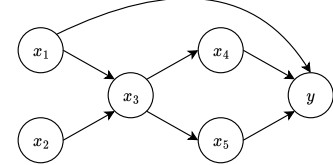
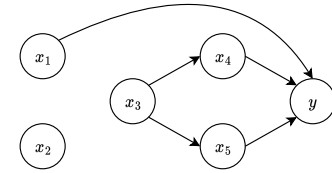

(a) Representation of an SCM node. Its value $x$ depends on the mapping function $f(C)$, where $C$ are the node's parents.

(b) An example of DAG without interventions.

(c) An example of an intervened DAG where the value of $x_3$ is not defined by the cause-effect relationship.

Figure 1: Representation of a node in a causal graph and how interventions are included to the feature $x_3$.

neural networks, directly fits the models to the distribution of the parents' values on target values, ensuring explicit causal propagation along the graph. We opt not to use random tree-based models, like other generators (Hollmann et al., 2025; Qu et al., 2025), due to the risk of having splits that are outside the parents' distribution, which could lead to small variation in the underlying causal chain or single-class outputs. This also allows us to have more explicit shift events. For detailed information about the target functions, refer to Appendix B.

After initialization, the generator is ready to produce data samples by propagating values through the graph, respecting the learned mappings and underlying causal dependencies. CaDrift can generate a large variety of datasets that propagate cause-and-effect relationships between features and the target without the need for a source dataset, like other generators do (Lin et al., 2024). Thus, the generated datasets do not violate constraints of data privacy or leakage. Both classification and regression tasks can be generated by CaDrift, depending on the target node chosen. In addition, unlike other SCM-based synthetic generators, CaDrift introduces time dependence into feature values.

**Interventions to Simulate Perturbations.** We leverage the concept of interventions in causal modeling (Pearl, 2010) to simulate environmental perturbations. Interventions are applied by forcing values to specific features without accounting for the cause-and-effect relations from their parent nodes, mimicking real-world perturbations such as equipment failures, environmental shocks, or deliberate overrides.

In practice, this is achieved by ignoring all of the edges that reach the intervened node, and a value is attributed to the node regardless of its mapping function, as in Figure 1c, where the intervened feature is $x_3$. Therefore, the value of the intervened node is not measured by its usual effect function $f_{x_3}(x_1, x_2)$. Instead, CaDrift forces values to intervene in features based on Normal or Uniform distributions in the case of continuous features, and random categories for categorical features.

The effect in the causal chain after this intervention can be described using do-calculus, as introduced by Pearl (1995). The resulting distribution of the label node after the intervention on $x_3$ is denoted as $P(y|\text{do}(x_3))$, where the do notation refers to an intervention. For a Normal distribution, we can write this as $P(y|\text{do}(x_3 \sim \mathcal{N}(\mu, \sigma^2)))$. By incorporating such interventions in a small proportion of the generated examples, we introduce occasional deviations, thus simulating noise and perturbations.

**Introducing Shifts.** To simulate concept shifts, CaDrift modifies mapping functions between nodes, thus modifying causal relationships. Modification in a single node affects the causal chain of its descendants in the graph. This way, it is possible to induce various types of shifts. Below, we describe how common types of shifts are induced in CaDrift:

- **Distributional Shift:** Simulated by changing mapping functions in the edges between nodes and by drifting the node mapper of the target label. Changing mapping functions alters how one feature affects another, which in turn modifies the downstream causal chain. When modifying the target node mapper, we are drifting $f_y(C)$, causing it to change the posterior probability $P(y|X)$, given that features $X$ are causes, direct or indirect, of $y$.

- **Covariate Shift:** To change the data distribution $P(X)$ in the input space, we change the parameters of the Normal/Uniform distributions used to generate the values for the root nodes in the causal graph. Changes in parameters of root nodes do not affect cause-and-

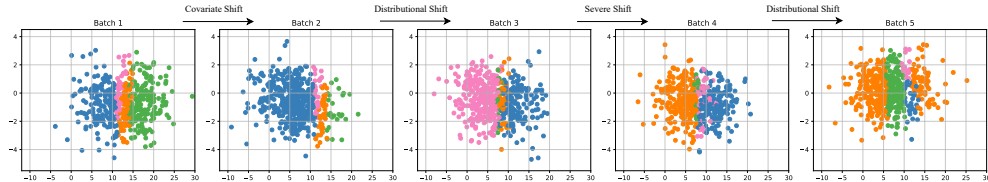

Figure 2: Samples generated by CaDrift using a DAG with six nodes – five features and one target. Each color refers to a different class.

effect relationships $f_E(C)$, but induce them to move to a different area in the feature space, which is propagated to the downstream nodes.

- **Severe Shift:** Simulated by inverting the outcome of the mapper function in the output between two different classes, i.e., changing the outcome of $f_y(C)$.

- **Local Shift:** This is a subtype of *covariate shift*, which happens when the distribution of the input space of a single feature changes. It can be generated using the same strategy as in the *covariate shift*, but affecting a single feature.

In addition, by considering the rate of change, we introduce a parameter $\Delta$ that defines the length of the drift window, allowing for abrupt, gradual, and incremental shifts. *Abrupt shift* happens in one step in time, thus, $\Delta = 1$. Gradual and incremental, on the other hand, have $\Delta > 1$. In a *gradual shift*, two concepts coexist during $\Delta$ time steps, while in the *incremental shift*, there are small steps in distribution at each arrived instance starting at time $t$, until the new concept is completely established at time $t + \Delta$. We can also simulate *recurrent concept shift*, where an old state of nodes in the graph is retrieved, thus returning to an old concept.

In Figure 2 we show batches generated by CaDrift using the DAG shown in Figure 1b. The features in the $x$ and $y$ axes are $x_1$ and $x_4$, two parents of the target $y$. The cause-and-effect relationship functions of this graph, by omitting the noise terms, can be written as:

- $x_1 \sim \mathcal{N}(\mu, \sigma^2)$
- $x_2 \sim \mathcal{U}(a, b)$
- $x_3 = f_{x_3}(x_1, x_2)$
- $x_4 = f_{x_4}(x_3)$
- $x_5 = f_{x_5}(x_3)$
- $y = f_y(x_3, x_4, x_5)$

There is a concept shift between each of the batches. For simplification purposes, only *abrupt shifts* are considered in this example. Specific details about mapping and target functions and how each shift is introduced in the stream, with more examples of class distributions generated by CaDrift, can be found in Appendix I.

From batch 1 to batch 2, we notice that the covariate shift did not affect class distribution, but the feature space moved towards a different area, as expected. From batch 2 to batch 3, the distributional shift has affected the decision boundary significantly, as well as the severe shift from batch 3 to 4, where we see clearly that two classes have swapped. Finally, the distributional shift that occurs in batch 5 also clearly affects the decision boundary. Even though this is a low-dimensional example, it offers us a visualization of CaDrift's power in generating controllable shift events and affecting data distribution and class boundaries. CaDrift has been made available on GitHub[2].

## 5 EXPERIMENTS

**Experimental Setup** The experiments in this paper are divided into two parts: 1) performance evaluation, to which we assess the performance of ML models on classifying data streams generated, and 2) stationarity tests, where we assess CaDrift's capability on generating non-iid samples.

The data stream baselines and the hyperparameters used in the experiments are presented in Appendix J. To run the data stream baselines, we use the (Bifet et al., 2010a) framework We apply

---

[2]Source code available on supplementary material during revision.

the same implementation to adapt TabPFN (Hollmann et al., 2025) for data streams as described by Lourenço et al. (2025), running on an NVIDIA A6000 GPU. All of the results reported are an average of 5 runs.

## 5.1 THE IMPACT OF SHIFT EVENTS

We begin by experimenting with datasets generated from the sample DAG shown in Figure 1b (datasets 1-3). Dataset 1 corresponds to the same example presented in Figure 2. Using the same DAG, we construct two additional datasets by varying the mapping functions and the drift events. A detailed description of the DAGs used in these experiments is provided in Appendix I, and the class distribution for datasets 2 and 3 is presented in Figure 8, also in Appendix I.

To assess performance on more complex settings, datasets 4 and 5 are generated with 10 and 25 features, respectively. Finally, datasets 6-8 are derived from larger graphs with 100-200 nodes, from which we randomly subsample features to form the final datasets. This subsampling emulates real-world scenarios where not all causal factors are observable or measurable (e.g., we do not know every variable that contributes to cancer development or market fluctuations). Information such as sample size, number of classes, balancing, etc., can be found in Table 9 (Appendix I). We also run experiments on popular synthetic generators for drifting data streams: SEA (Street & Kim, 2001), Sine (Gama et al., 2004), and RandomRBF (Bifet et al., 2009b), to which 10,000 instances were sampled using the river library (Montiel et al., 2021). On the SEA and Sine datasets, drift events happen every 2,500 instances. On RandomRBF, there is an incremental drift that persists throughout the whole stream, induced by, at each step, changing the position of the centroids.

Table 2 shows the average accuracy and average rank on the baselines. The baselines' prequential accuracy on the generated samples is shown in Figure 3. The sliding window size used to calculate the prequential accuracy and the initial training size of baselines is set to 100 on datasets 1-6. For datasets 7 and 8 (100,000 instances), we set the size of the prequential accuracy window to 1,000 in order to obtain a smoother prequential curve (Bifet et al., 2015).

Table 2: Average accuracy and standard deviation of baselines on datasets generated by CaDrift and popular synthetic benchmarks.

| Dataset | IncA-DES | TabPFN$^{Stream}$ | ARF | LevBag | OAUE | HT | LAST |
|---|---|---|---|---|---|---|---|
| 1 | 86.69 ±0.17 | 68.83 ±0.13 | **87.78** ±0.18 | 78.08 ±1.59 | 62.38 ±0.00 | 59.08 ±0.00 | 86.17 ±0.00 |
| 2 | 70.73 ±1.17 | 67.18 ±0.22 | **73.82** ±0.43 | 72.94 ±0.80 | 45.33 ±0.00 | 53.00 ±0.00 | 67.67 ±0.00 |
| 3 | **87.33** ±0.11 | 75.44 ±0.07 | 84.50 ±0.23 | 82.23 ±0.20 | 75.04 ±0.00 | 76.92 ±0.00 | 78.21 ±0.00 |
| 4 | 67.49 ±0.27 | **86.00** ±0.16 | 66.99 ±0.09 | 65.06 ±2.24 | 67.23 ±0.00 | 45.51 ±0.00 | 66.43 ±0.00 |
| 5 | 91.61 ±0.12 | **96.85** ±0.06 | 94.91 ±0.03 | 94.47 ±0.15 | 88.99 ±0.00 | 88.08 ±0.00 | 91.17 ±0.00 |
| 6 | 73.80 ±0.30 | **80.26** ±0.08 | 76.75 ±0.13 | 75.17 ±0.21 | 72.36 ±0.00 | 66.23 ±0.00 | 69.58 ±0.00 |
| 7 | 32.49 ±0.21 | 35.93 ±0.12 | 35.96 ±0.02 | **35.99** ±0.03 | 35.75 ±0.00 | 34.68 ±0.00 | 34.48 ±0.00 |
| 8 | 74.91 ±0.20 | 78.94 ±0.02 | 79.01 ±0.05 | **79.58** ±0.03 | 79.44 ±0.00 | 77.63 ±0.00 | 77.24 ±0.00 |
| SEA | 96.55 ±0.14 | **97.42** ±0.04 | 96.56 ±0.10 | 95.61 ±0.32 | 92.95 ±0.00 | 91.30 ±0.00 | 91.61 ±0.00 |
| Sine | **96.32** ±0.03 | 85.77 ±0.05 | 95.79 ±0.06 | 87.75 ±1.40 | 79.92 ±0.00 | 52.92 ±0.00 | 89.81 ±0.00 |
| RandomRBF | 62.44 ±0.14 | **65.77** ±0.06 | 64.20 ±0.16 | 55.07 ±0.44 | 51.05 ±0.00 | 51.82 ±0.00 | 51.82 ±0.00 |
| Average | 76.40 | 76.22 | **77.84** | 74.90 | 68.22 | 63.38 | 73.11 |
| Av. Rank | 3.4 | 3.0 | **2.2** | 3.0 | 5.3 | 6.2 | 4.9 |

On datasets 1 and 2, the *covariate shift* introduced at the 500th instance produces no visible drop in classifier performance – an expected behavior, as *covariate shift* preserves causal relationships between nodes while shifting the regions of the feature space being sampled. We can also observe that methods that incorporate some adaptation strategy, such as ARF and LevBag, exhibit better resilience to distributional shifts than HT and even TabPFN$^{Stream}$, which carries no adaptation strategy.

Interestingly, the performance of TabPFN is better for the first two concepts; however, its classification performance drops after the first distributional shift. This happens because TabPFN's context window is set to 1,000 instances, while concepts in datasets 1-3 lasted for 500 instances. Hence, TabPFN had instances from two different concepts in its context window. This could be easily contoured by defining a smaller context sliding window. However, in real-world applications, concept duration is usually unknown. Bigger sliding windows facilitate learning of stable concepts, while smaller ones lead to quicker adaptation, a phenomenon known as the stability-plasticity dilemma (Mermillod et al., 2013).

On the incremental and gradual shift events in Figure 3b, we observe drops in performance like in *abrupt shifts*. The learning, primarily on the incremental change between instances 1,000 and

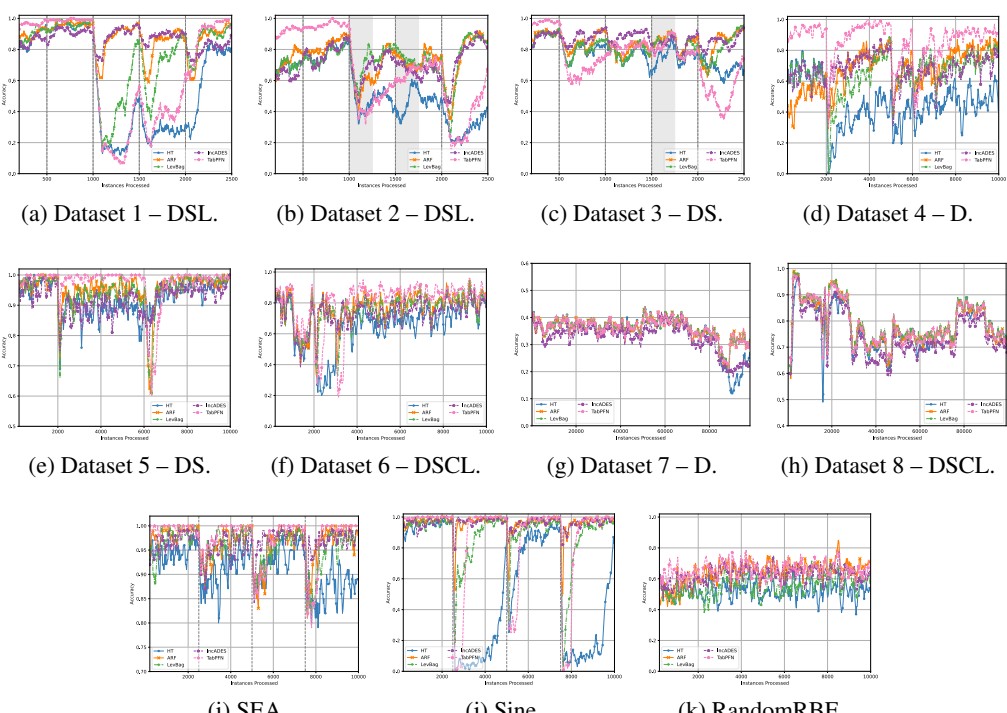

(a) Dataset 1 – DSL.  (b) Dataset 2 – DSL.  (c) Dataset 3 – DS.  (d) Dataset 4 – D.

(e) Dataset 5 – DS.  (f) Dataset 6 – DSCL.  (g) Dataset 7 – D.  (h) Dataset 8 – DSCL.

(i) SEA.  (j) Sine.  (k) RandomRBF.

Figure 3: Prequential accuracies on tested datasets. Dashed vertical lines indicate shift points. Shaded areas refer to the length of incremental and gradual shifts. Letters refer to the distributional shifts applied to the datasets. D stands for distributional, S for severe, C for covariate, and L for local shifts.

1,250, appears to be compromised due to the still-changing concept; however, after the concept is established, the performance curve exhibits a sharper increase. The behavior is slightly different during the gradual shift between instances 1,500 and 1,750, where variations in accuracy are observed during the length of the shift, and again with a sharp increase once it is established. HT, on the other hand, after a decline in accuracy, presents an increase in performance while the shift is still in progress.

In Figure 3c, we also notice drops in performance on shift events, even on the recurrent shift on the 1,000th data point. IncA-DES, which preserves information from previous concepts (Barboza et al., 2025), offers improved recovery from the recurrent shift. The gradual change in this dataset exhibits different behavior, characterized by a slight drop in accuracy over the course of the shift. Still, after the concept is established, we notice a sharp drop in performance. This suggests that learners were unable to properly grasp the new concept before it was fully integrated into the stream.

Getting into datasets with higher dimensionality, where we include random shift events with varying rates of change (abrupt, incremental, gradual, recurrent) in specific points in Figures 3d, 3e, 3f, 3g, and 3h we also notice that simulated concept shifts stress classifiers and require them to adapt. The drift events are also diverse, depending on the strategy used to induce them, meaning that some might not compromise the performance of classifiers, while others need more severe adaptation.

Classifiers tend to quickly learn the decision functions of popular baselines (SEA and Sine), as shown in Figures 3i and 3j, where they often reach 100% accuracy, despite drift events also requiring adaptability. The RandomRBF generator appears to be more challenging for the baselines, but drift events are limited to the moving of centroids. In contrast, CaDrift poses greater and more diverse challenges. Classifiers that fail to capture the causal relationships between features and the target struggle to achieve competitive performance. Moreover, each dataset highlights different aspects of classifier behavior, revealing both strengths and weaknesses. Additional experiments with limited label availability are available in Appendix E, and for regression tasks in Appendix H.

Overall, these results confirm that concept drift events generated by CaDrift pose challenges to classifiers, necessitating the implementation of proper adaptation strategies. Otherwise, their performance can be compromised. These features make CaDrift suitable for testing a wide range of time-dependent ML models and adaptive strategies.

## 5.2 STATIONARITY TESTS

In this section, we perform stationarity tests to assess how each component impacts the induction of serial correlation upon samples generated by CaDrift. In Table 3 we show the results of the Ljung-box test (Ljung & Box, 1978) on data stream sampled by CaDrift datasets without shift events, with five features each, in the form of an ablation study, where we test each component separately (EWMA and autoregressive noise, AR). Notice that the datasets' features without any mechanism to induce time dependency, i.e., the iid column, do not reject the $H_0$ of the test. This behavior should be similar to other SCM synthetic generators (Hollmann et al., 2025; Qu et al., 2025), as there is no mechanism to induce time dependence. When including EWMA ($\alpha = 0.05$), most of the features reject the null hypothesis, i.e., the values that features assume have serial correlation.

When using the autoregressive noise (AR) with $\rho = 0.1$, all of the features reject the null hypothesis. Thus, even with a small value for $\rho$, the autoregressive noise induces serial correlation in the features, which propagates to the target node. By merging both components, the Ljung-Box test confirms the presence of serial correlation in every feature and target, the exact same behavior observed on real-world datasets (see Appendix F.1). We conduct the same Ljung-box test on popular synthetic data stream generators in Appendix F.1, and confirm that instances sampled by these generators have no serial correlation.

Table 3: Ljung-Box test (20 lags) on synthetic data streams generated by CaDrift with different strategies for time dependence.

|    | iid | | EWMA | | AR | | EWMA+AR | |
|----|---------|-------------|---------|-------------|---------|-------------|---------|-------------|
|    | p-value | Reject $H_0$ | p-value | Reject $H_0$ | p-value | Reject $H_0$ | p-value | Reject $H_0$ |
| x1 | 0.194 | N | < 0.001 | Y | < 0.001 | Y | < 0.001 | Y |
| x2 | 0.444 | N | < 0.001 | Y | < 0.001 | Y | < 0.001 | Y |
| x3 | 0.716 | N | < 0.001 | Y | < 0.001 | Y | < 0.001 | Y |
| x4 | 0.412 | N | < 0.001 | Y | < 0.001 | Y | < 0.001 | Y |
| x5 | 0.386 | N | 0.612 | N | < 0.001 | Y | < 0.001 | Y |
| y  | 0.076 | N | 0.094 | N | 0.01 | Y | < 0.001 | Y |

Complementary experiments, including the impact of the $\alpha$ parameter on EWMA and Autocorrelation Function (ACF) plots, can be found in Appendix F. In summary, the autoregressive noise induces more substantial serial autocorrelation, while EWMA serves as a smoothing factor for the values assigned to root nodes in the stream, in addition to allowing samples to carry memory from past instances. The serial correlation propagates through the causal chain until the target node, as we can observe in Figure 5 in Appendix F. We also report experiments using the Maximum Mean Discrepancy (MMD) (Gretton et al., 2012) between joint distributions in Appendix G. These results show that CaDrift produces temporally evolving, non-stationary distributions that more closely resemble the behavior of real-world streams than existing synthetic benchmarks.

## 6 CONCLUSION

In this work, we have presented CaDrift, a causal framework to generate synthetic time-dependent tabular data with concept drift. CaDrift provides controllable shift events that affect the performance of classifiers. CaDrift's flexibility allows the synthesis of distributional, covariate, severe, and local shifts that may occur at different rate changes, including abrupt, gradual, incremental, and recurrent settings. Moreover, we confirm that samples generated by CaDrift have serial correlation through statistical tests, making it a valuable tool to create and evaluate models under evolving data.

Future work includes using the presented framework to replicate causal relationships found in real-world datasets, thereby introducing shifts in them. Additionally, samples generated by CaDrift can also work as prior for the training of time-dependent tabular foundation models (van Breugel & van der Schaar, 2024). We hope this generator supports practitioners and researchers in more effectively assessing and exploring the challenges and opportunities of drifting data streams.

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

## A   MAPPING FUNCTIONS

The mapping functions are used to map the values of nodes based on their parents. For the root nodes, the mapping functions utilized are based on the Normal and Uniform distributions:

- Normal: $\mathcal{N}(\mu, \sigma^2)$
- Uniform: $\mathcal{U}(a, b)$

Each root node is mapped by one of these two functions, chosen randomly. The parameters ($\mu$, $a$, $b$, and $\sigma$) are randomly initialized. For the inner nodes, the mapping functions are ML models trained to approximate various labeling functions, such as linear, step, or sine, as well as a multilayer perceptron (MLP) with random weights initialized using Xavier's initialization (Glorot & Bengio, 2010), as also done in TabPFN's generator (Hollmann et al., 2025). The models considered to map values of inner nodes are:

- Learned MLP.
- Random MLP.
- Decision Tree.
- Linear regression model optimized through stochastic gradient descent.

Each of these mappers (except the random MLP) assigned to a node learn a target function from the parent nodes. We use a linear regression model optimized via stochastic gradient descent to allow incremental updates in node values, and thus simulate incremental drift. Thus, these models aim at a regression mapping for the nodes' values. We chose not to use random tree models, such as other SCM generators (Hollmann et al., 2025), due to the potential for splits that fall outside the parents' distribution, which increases the risk of negatively affecting the causal chain, leading to small variations or single-class outputs. By using target functions, the shifts are also more explicit.

Table 4: Hyperparameters of mapping functions that map cause-and-effect relationships in the SCM. All mappers were implemented with `scikit-learn`).

| Mapper | Hyperparameters |
|---|---|
| Learned MLP | $hidden\_layers = 1$, $neurons = 10$, $optimizer = adam$, $max\_iter = 10$, $activation = relu$, $learning\_rate = 0.001$ |
| Random MLP | $hidden\_layers = 1$, $neurons = 10$ |
| Decision Tree | $max\_depth \in [5, 25]$, $criterion =' squared\_error'$ |
| Regression w/ SGD | $max\_iter = 10$, $penalty = l2$, $alpha = 0.0001$ |

To induce concept drift on the mappers, we employ different strategies depending on the mapper: refitting the mapper on a different target function; the weights of the Random MLP are reinitialized; the linear regression, as it can be trained incrementally, can also be induced incremental concept drift by, at each time step, fitting the model to an instance sampled with a new target function. The mappers for categorical features are:

- Categorical Prototype Mapper (Hämäläinen et al., 2017).
- Gaussian Prototype Mapper (Rasmussen, 1999).
- Random Radial Basis Function (Bifet et al., 2009b).
- Rotating Hyperplane (Hulten et al., 2001).

Most of these mappers assign the category based on the proximity of parents' values to centroids/prototypes. The prototypes and centroids are initialized randomly based on the distribution of the parent nodes during the initialization process. To introduce concept shift in these functions, we also employ different strategies: 1) change the prototypes positions; 2) change the distance function of the Categorical Prototype Mapper; 3) small step in prototypes' position in each time step to induce incremental shift; 4) Rotating the hyperplane in incremental steps or suddenly; 5) shift the outcome of two different classes in order to induce severe shift. A detail regarding our categorical mappers implementation is that a class can be assigned to more than one centroid, which provides more complex and diverse decision boundaries.

## B  TARGET FUNCTIONS

The target functions are those that the mappers learn to map according to the parents' values. Rather than simply choosing a target function to map the values, having ML models to learn them intro-

duces more complexity to the cause-and-effect relationships and includes approximation errors. The functions can be chosen either randomly or manually for each inner node, and are listed below:

- Linear function: $f(X) = \sum_{i=1}^{d} w_i x_i + b + \epsilon$

- Sine function: $f(X) = \sum_{i=1}^{d} \sin x_i + \epsilon$

- Step function: $f(X) = \begin{cases} 1 + \epsilon, & \text{if } \sum_{i=1}^{d} x_i > 0 \\ 0 + \epsilon, & \text{otherwise} \end{cases}$

- Checkerboard function: $f(X) = \sum_{i=1}^{d} \lfloor x_i \rfloor \mod 2$

- Radial Basis Function: $f(X) = \exp\left(-\frac{||X||^2}{2\sigma^2}\right) + \epsilon$

## C  CaDrift Pseudocode

The pseudocode for generating data samples with CaDrift is exposed in Algorithm 1, and for sampling a DAG in Algorithm 2. The algorithm receives as parameters the probability $p_i$ that samples receive an intervention, the probability $p_m$ that samples have missing features, the dimensionality $d$, and the minimum and maximum number of parents for each node. The algorithm starts by sampling a DAG with $d+1$ nodes (dimensionality and target node). In Line 2, an empty list for the samples is initialized. For each sample to be generated, first, it is checked if there will be any intervened node or missing feature in Lines 6-10.

The sample generation follows the topological order of the graph, starting from root nodes and traversing to their descendants. For root nodes, the values are mapped through a Normal or Uniform distribution; thus, there are no parents to map to (Line 16). For inner nodes, the values are computed according to the effect function $f_E$, which maps cause-and-effect relationships from parents to the node (Line 18). After traversing the graph and computing the values for each node, the generated instance is added to the list in Line 23, which is returned in Line 26.

The algorithm to build a DAG (Algorithm 2) starts by initializing a graph $\mathcal{G}$ with $d + 1$ nodes (including the target) with $n\_roots$ nodes as roots. The root nodes are randomly assigned to either a Normal or Uniform mapper in Line 4. For each inner node, the number of parents is chosen randomly in the range $[min\_parents, max\_parents]$. Random parents are chosen in Line 7, and the edges are added to each node in Lines 9-10. To each inner node, there is attributed one of the mapping functions described in Appendix A and one of the target functions described in Appendix B in Lines 12-13. Finally, a random categorical mapper is chosen to work as the target variable $y$.

## D  The impact of the $\alpha$ parameter in EWMA

Let us assess the evolution of EWMA with different $\alpha$ values along with the autoregressive noise with $\rho = 0.5$, shown in Figure 4. Notice that the EWMA acts as a smoothing factor, in which smaller values of $\alpha$ lead to a smoother evolution of the average over time, while higher values give more weight to recent values, and thus follow the noise more closely. Hence, the combination of the autoregressive structure of the generation process and the effect of EWMA results in a smooth evolution of the values that propagate through the causal chain over time.

## E  Results with partial label availability

We present results on the datasets used in experiments with partial label availability and a delay in label arrival. These experiments facilitate the evaluation of methods in an environment closer to those found in the real world, where ground-truth labels are not readily available. We employ a delay of 100 instances, and 1 every 2 instances arriving in the stream are labeled (i.e., 50% of samples are labeled). Results are in Table 5, where values in parentheses refer to the difference in average accuracy to the test-then-train policy.

HT is the method that presented the smallest drop in the accuracy – also the one with the smallest accuracy in test-then-train. In contrast, TabPFN$^{\text{Stream}}$ had the highest drop, with a difference of

---

**Algorithm 1** Generate Data ($dataset\_size$, $p_i$, $p_m$, $d$, $min\_parents$, $max\_parents$ )

---

1: $Graph\ \mathcal{G} \leftarrow$ Build DAG($d, min\_parents, max\_parents$)
2: Initialize $samples[node] \leftarrow []$ for each $node \in \mathcal{G}$
3: **for** $n = 0$ **to** $dataset\_size$ **do**
4:     $intervened\_nodes \leftarrow \emptyset$
5:     $missing\_nodes \leftarrow \emptyset$
6:     **if** random() $< p_i$ **then**
7:         Select 1–3 random nodes as $intervened\_nodes$
8:     **end if**
9:     **if** random() $< p_m$ **then**
10:        Select 1–3 random nodes as $missing\_nodes$
11:     **end if**
12:     **for** each $node$ in $\mathcal{G}.topological\_order()$ **do**
13:        **if** $node \in intervened\_nodes$ **then**
14:           Apply intervention to $node$
15:        **else if** $node$ is root **then**
16:           $node.value \leftarrow compute\_value()$
17:        **else**
18:           $node.value \leftarrow compute\_value(node.parents)$
19:        **end if**
20:        Append $node.value$ to $samples[node]$
21:     **end for**
22:     **for** each $node \in missing\_nodes$ **do**
23:        $samples[node][n] \leftarrow$ NaN
24:     **end for**
25: **end for**
26: **return** $samples$

---

---

**Algorithm 2** Build DAG ($d$, `n_roots`, `min_parents`, `max_parents`)

---

1: initialize graph $\mathcal{G}$ with nodes $V \in x_1, x_2, \dots x_{d+1}$
2: `root_nodes` $\leftarrow$ randomly select `n_roots` nodes as roots
3: **for** node $\in$ `root_nodes` **do**
4:     Randomly assign Normal or Uniform distribution to node
5: **end for**
6: **for** each non-root $node \in$ `topological_order` **do**
7:     $choose\ num\_parents \in [min\_parents, max\_parents]$
8:     select $parents$ from $\{x_1, \dots, x_{i-1}\}$
9:     **for** each $parent$ in $parents$ **do**
10:        add edge from $parent$ to $node$
11:     **end for**
12:     $choose$ mapping function
13:     $choose$ target function
14: **end for**
15: choose random `node` with a categorical mapper to be the label $y$
16: **return** $\mathcal{G}$

---

16.78 percentage points compared to the test-then-train policy. We perceive drops in accuracy on both popular synthetic generators (SEA, Sine, and RandomRBF), as well as on the data streams generated by CaDrift.

## F   EXTENDED STATIONARITY TESTS

In Figure 5, we show the ACF Plots of two features (root and inner nodes) and the target of datasets generated by the sample DAG in Figure 1b, with different values for the parameters $\alpha$ of the EWMA,

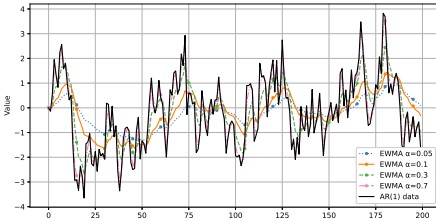

Figure 4: EWMA evolution with different $\alpha$ values. The lines show the raw values generated with autoregressive noise, and the impact of EWMA on the value depending on the parameter $\alpha$ assigned.

Table 5: Average accuracy of baselines with limited label availability. Values in parentheses refer to the difference in accuracy compared to the experiments performed in a test-then-train manner.

| Dataset | IncA-DES | TabPFN$^{Stream}$ | ARF | LevBag | OAUE | HT | LAST |
|---------|----------|-------------------|-----|--------|------|-----|------|
| 1 | **77.50**(9.19) | 43.71(25.12) | 75.54(12.24) | 64.88(13.20) | 27.63(34.75) | 55.67(3.41) | 74.25(11.92) |
| 2 | 61.25(9.48) | 52.33(14.85) | **61.29**(12.53) | 58.00(14.94) | 21.75(23.58) | 49.17(3.83) | 54.79(12.88) |
| 3 | **79.08**(8.25) | 74.29(1.15) | 77.29(7.21) | 76.17(6.06) | 75.20(-0.16) | 75.96(0.96) | 75.96(2.25) |
| 4 | **61.75**(5.74) | 29.98(56.02) | 57.23(9.06) | 59.05(8.01) | 56.45(10.78) | 42.70(2.81) | 63.86(2.57) |
| 5 | 90.04(1.57) | 72.37(24.48) | **92.63**(2.28) | 91.11(3.36) | 81.61(7.38) | 83.49(4.59) | 87.36(3.81) |
| 6 | 71.11(2.69) | **76.51**(3.75) | 72.94(3.81) | 71.19(3.98) | 67.18(5.18) | 64.33(1.90) | 67.10(2.48) |
| 7 | 33.79(-1.30) | 29.91(6.02) | **35.86**(0.10) | 35.81(0.18) | 35.46(0.29) | 34.56(0.12) | 34.31(0.17) |
| 8 | 73.81(1.10) | 77.09(1.85) | 78.60(0.41) | **79.21**(0.37) | 78.80(0.64) | 77.06(0.57) | 77.05(0.19) |
| SEA | 93.25(3.30) | 92.61(4.81) | **93.86**(2.70) | 93.24(2.37) | 87.06(5.89) | 91.86(-0.56) | 91.68(-0.07) |
| Sine | 78.23(18.09) | 49.44(36.33) | **90.26**(5.53) | 81.66(6.09) | 67.24(12.68) | 58.27(-5.35) | 85.21(4.60) |
| RandomRBF | 57.94(4.50) | **63.88**(1.89) | 57.22(6.98) | 51.93(3.14) | 49.04(2.01) | 51.34(0.48) | 51.34(0.48) |
| Average | 70.70(5.69) | 60.19(16.02) | **72.13**(5.71) | 69.30(5.61) | 58.86(9.37) | 62.22(1.16) | 69.36(3.75) |
| Av. Rank | 3.09 | 5.09 | **1.91** | 2.82 | 5.45 | 5.27 | 4.18 |

and the $\rho$ of the autoregressive noise. These plots illustrate the serial correlation of values explicitly, and the effect propagates to the target node, resulting in autocorrelation in the labels.

Notice that higher values for $\rho$ tend to increase the autocorrelation of both features and target on early lags, and the autocorrelation gradually decreases as the lag increases – which shows that consecutive data samples are dependent on each other. When $\rho = 0$, we see that the autocorrelation on every lag and $\alpha$ value is negligible.

## F.1 LJUNG-BOX TEST ON SYNTHETIC GENERATORS AND REAL-WORLD DATASETS

We conduct the Ljung-box test on popular synthetic data stream generators: RandomRBF (Bifet et al., 2009b), SEAConcepts (Street & Kim, 2001), and Sine (Gama et al., 2004). Results are shown in Table 6, and we confirm that popular data stream generators have no serial correlation in either the features or target. Despite drift events, generated samples are iid. The exception is RandomRBF, which presents serial correlation on some features, but that does not propagate to the interest variable. When comparing to the real-world datasets (Table 7), samples generated by CaDrift are more aligned with real scenarios, where we notice serial correlation on all features, including the interest variable $y$.

Table 6: Ljung-Box test (20 lags) on synthetic data streams generated by popular synthetic generators for data streams.

| | RandomRBF | | SEAConcepts | | Sine | |
|---|---|---|---|---|---|---|
| | p-value | Reject $H_0$ | p-value | Reject $H_0$ | p-value | Reject $H_0$ |
| x1 | 0.011 | Y | 0.299 | N | 0.387 | N |
| x2 | 0.012 | Y | 0.626 | N | 0.324 | N |
| x3 | 0.192 | N | 0.337 | N | – | – |
| x4 | 0.612 | N | – | – | – | – |
| y | 0.723 | N | 0.083 | N | 0.547 | N |

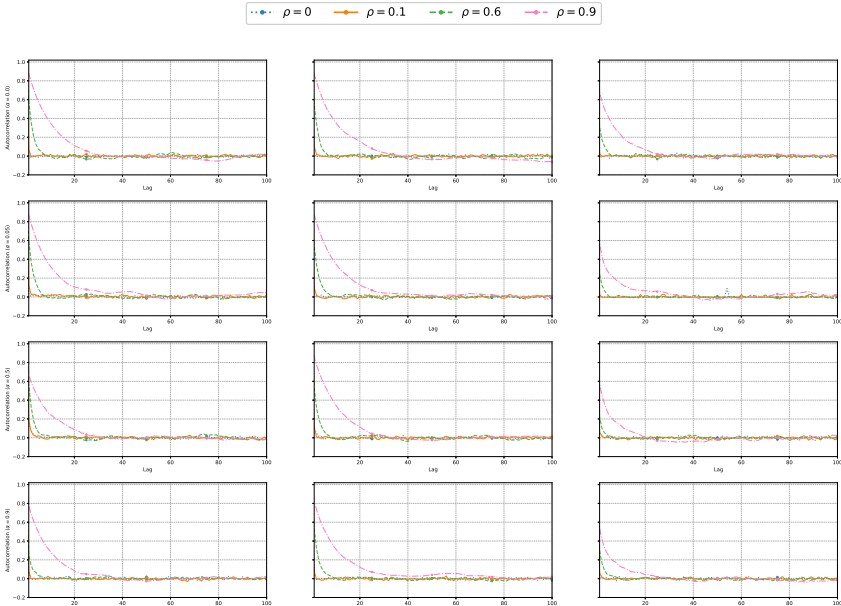

Figure 5: The impact of the $\alpha$ and $\rho$ variables on the lagged autocorrelation function. Each row refers to a different value for $\alpha$, and each column a different feature ($x_1$, $x_3$ and $y$).

Table 7: Ljung-Box test (20 lags) on real-world datasets.

|  | Electricity | | NOAA | |
|---|---|---|---|---|
|  | p-value | Reject $H_0$ | p-value | Reject $H_0$ |
| x1 | < 0.001 | Y | < 0.001 | Y |
| x2 | < 0.001 | Y | < 0.001 | Y |
| x3 | < 0.001 | Y | < 0.001 | Y |
| x4 | < 0.001 | Y | < 0.001 | Y |
| x5 | < 0.001 | Y | < 0.001 | Y |
| x6 | < 0.001 | Y | < 0.001 | Y |
| x7 | < 0.001 | Y | < 0.001 | Y |
| x8 | < 0.001 | Y | < 0.001 | Y |
| y | < 0.001 | Y | < 0.001 | Y |

## G    DISTRIBUTION DISTANCES

In Figure 6, we report the Maximum Mean Discrepancy (MMD) computed on the joint distribution $P(X, y)$ for both synthetic and real-world datasets. The datasets generated by CaDrift (Figures 6a–6h) exhibit substantial variation in MMD over time, with significant discrepancies between batches at different points in the stream. This behavior closely mirrors that of real-world datasets (Figures 6l and 6m). In contrast, popular synthetic generators (Figures 6i–6k) generally show MMD values below 0.1, with the exception of the Sine dataset, which reaches approximately 0.2. These results indicate that CaDrift produces data whose distributions evolve more substantially over time, whereas existing synthetic benchmarks, despite allowing predefined shifts, induce comparatively small distributional changes across the stream. In this sense, CaDrift more closely reflects the temporal distributional variability observed in real-world data.

## H    REGRESSION TASKS

In Figure 7, we report the Mean Absolute Error (MAE) of baseline regressors available in the River library (Montiel et al., 2021) on streams generated by CaDrift using the sample DAG in Figure 1b. Shift events occur every 2,000 instances, and we observe that these events lead to clear changes in

model performance, indicating that regressors must re-adapt to the evolving data distribution. In this setting, *covariate shifts* also manifest in the error dynamics, as evidenced by the variations in MAE immediately following each shift. These results demonstrate that CaDrift can naturally generate time-evolving challenges for regression tasks.

# I    INFORMATION OF GENERATED DATASETS

The dataset presented in Figure 2 contains events of covariate shift, distributional shift, and severe shift. Table 8 exposes the information of each node. Furthermore, in the experiments in Section 5.1, we generate two more datasets using the same DAG, but with different mapping functions and shift events, and also describe them in Table 8.

Table 8: Information of Mappers and Target Functions of nodes in the sample DAG. Normal and Uniform distributions carry no target function, as well as the random MLP and the prototype-based categorical mapper.

| Node | Mapper | Target Function |
|---|---|---|
| *Dataset 1* | | |
| $x_1$ | Normal Distribution | – |
| $x_2$ | Uniform Distribution | – |
| $x_3$ | MLP | Sine |
| $x_4$ | Random MLP | – |
| $x_5$ | Linear Reg. | Checkerboard |
| $y$ | Prototype-based Categorical Mapper | – |
| *Dataset 2* | | |
| $x_1$ | Normal Distribution | – |
| $x_2$ | Uniform Distribution | – |
| $x_3$ | Decision Tree | Linear |
| $x_4$ | Random MLP | – |
| $x_5$ | Linear Reg. | Radial Basis |
| $y$ | RandomRBF Categorical Mapper | – |
| *Dataset 3* | | |
| $x_1$ | Normal Distribution | – |
| $x_2$ | Uniform Distribution | – |
| $x_3$ | Decision Tree | Step |
| $x_4$ | Random MLP | – |
| $x_5$ | Linear Reg. | Sine |
| $y$ | Gaussian Categorical Mapper | – |

Table 9 presents a summary of information regarding the datasets generated for the experiments in Section 5.1. They were generated using $\rho = 0.5$, a moderate persistence in the autoregressive noise, and $\alpha = 0.05$, which filters the autoregressive noise and makes consecutive samples slightly dependent on past values.

Table 9: Information on the datasets used in the experiments. Datasets were generated using $\alpha = 0.05$ and $\rho = 0.5$.

| Dataset | # dim. | # classes | # concepts | % min. class | # samples | % Missing | % Interventions |
|---|---|---|---|---|---|---|---|
| 1 | 5 | 4 | 5 | 16.48% | 2,500 | 0 | 0 |
| 2 | 5 | 5 | 5 | 8.76% | 2,500 | 0 | 0 |
| 3 | 5 | 3 | 5 | 10.4% | 2,500 | 0 | 0 |
| 4 | 10 | 10 | 10 | 1.98% | 10,000 | 0 | 10% |
| 5 | 25 | 2 | 10 | 33.53% | 10,000 | 10% | 10% |
| 6 | 100 | 3 | 20 | 13.46% | 10,000 | 10% | 10% |
| 7 | 10 | 7 | 20 | 0.21% | 100,000 | 0 | 10% |
| 8 | 25 | 2 | 100 | 30.90% | 100,000 | 0 | 10% |

Regarding the shift events, there were a total of 4 in each sample small-scale dataset, resulting in five different concepts, each with 500 instances, described in detail in Table 10.

Table 10: Information regarding shift events on the datasets used as examples in the paper. $\Delta$ refers to the shift length.

| Shift | Type | $\Delta$ | Description |
|---|---|---|---|
| *Dataset 1* | | | |
| 1 | Abrupt Covariate Shift | 1 | Mean of Normal distribution in $x_1$ changed. |
| 2 | Abrupt Distributional Shift | 1 | Position of centroids changed; Target function of node $x_5$ changed to a Sine Function. |
| 3 | Abrupt Severe Shift | 1 | The outcome between two classes was swapped. |
| 4 | Abrupt Distributional Shift | 1 | Position of centroids changed; Target function of node $x_3$ changed to a Step Function. |
| *Dataset 2* | | | |
| 1 | Abrupt Covariate Shift | 1 | Mean of Uniform distribution in $x_2$ changed. |
| 2 | Incremental Distributional Shift | 250 | Position of centroids slightly change at each time step; Target Function of node $x_5$ changed to a Checkerboard Function. |
| 3 | Gradual Severe Shift | 250 | The outcome between two classes was swapped. During drift length, both concepts are sent. |
| 4 | Abrupt Distributional Shift | 1 | Position of centroids changed; Target function of node $x_3$ changed to a Sine Function. |
| *Dataset 3* | | | |
| 1 | Abrupt Distributional Shift | 1 | Position of centroid changed; Random MLP of node 4 reinitialized. |
| 2 | Abrupt Recurrent Shift | 1 | State of past concept retrieved. |
| 3 | Gradual Severe Shift | 250 | The outcome between two classes was swapped. During the shift length, both concepts are sent together in the stream. |
| 4 | Abrupt Distributional Shift | 1 | Position of centroids changed; Target function of node $x_3$ changed to a Linear Function. |

The samples generated by the sample dataset 2 are shown in Figure 8a, and dataset 3 in Figure 8b. Regarding dataset 2, note that the class distributions change for distributional and severe shifts, with special attention to the severe shift that occurred in batch 4, where we observe some overlap between two classes – two concepts coexist while the drift window lasts. The incremental distributional shift in batch 3 appears to have affected the class distribution since the first step, as suggested by the small similarity to the previous concept. The action of shift events on the decision boundary varies a lot.

On dataset 3 (Figure 8b), distributional shift also changes the class distribution on the feature space. The recurrent shift from batch 2 to 3 makes the class distribution the same as in the first batch. A severe shift, similar to the other datasets, swaps the outcome between two classes. The last distributional shift, which occurs in batch 5, alters the class distribution of the classes.

## J   BASELINES

In this section, we describe the baseline methods used for evaluating CaDrift. A summary is available in Table 11. Below, we give details of each method:

- Hoeffding Tree (HT): An incremental learning that uses the Hoeffding bound to split the nodes of the tree (Domingos & Hulten, 2000).

- LAST: A tree-based incremental learner that uses drift detectors in the process of splitting nodes (Assis et al., 2025).

- LevBag: An incremental ensemble learning method for drifting data streams. The classifiers are trained using Online Bagging with $Poisson(\lambda = 6)$ (Bifet et al., 2010b).

- OAUE: An incremental ensemble method that maintains weighted classifiers. Weights are updated based on the predictive error on the most recent data block. It replaces the worst-performing component with a new classifier when a new data block is available (Brzezinski & Stefanowski, 2014).

- ARF: An adaptive ensemble method for drifting data streams. Each base classifier carries its own drift detector. When a single classifier becomes outdated (drift is detected), it is replaced by a new one trained on more recent samples. Classifiers are trained using Online Bagging with $Poisson(\lambda = 6)$ (Gomes et al., 2017).

- IncA-DES: A dynamic ensemble selection method for drifting data streams. Classifiers are trained using an incremental training approach. The region of competence is computed through an Online K-d tree. The best classifiers in the region of competence are selected to compose the ensemble (Barboza et al., 2025).

- TabPFN: A prior-fitted transformer-based method. It was pre-trained on millions of datasets generated through SCMs (Hollmann et al., 2025). We use the version adapted for data streams described by Lourenço et al. (2025).

We have used the same hyperparameters as in the MOA framework, and for IncA-DES, the same as in the original paper (Barboza et al., 2025). The hyperparameters are described in Table 12.

Table 11: Baseline methods for drifting data streams.

| Method | Source | Category |
|---|---|---|
| HT | Domingos & Hulten (2000) | Online Learner |
| LAST | Assis et al. (2025) | Online Learner |
| LevBag | Bifet et al. (2010b) | Ensemble |
| OAUE | Brzezinski & Stefanowski (2014) | Ensemble |
| ARF | Gomes et al. (2017) | Ensemble |
| IncA-DES | Barboza et al. (2025) | Dynamic Ensemble Selection |
| TabPFN[Stream] | Lourenço et al. (2025) | Transformer |

Table 12: Baselines' hyperparameters. All ensemble methods use a HT as base classifier.

| Method | Hyperparameters |
|---|---|
| HT | $grace\_period = 200$ |
| LAST | $change\_detector : ADWIN$ |
| LevBag | $change\_detector : ADWIN, ensemble\_size = 10$ |
| OAUE | $ensemble\_size = 10, window\_size = 500$ |
| ARF | $change\_detector : ADWIN, ensemble\_size = 100$ |
| IncA-DES | $change\_detector : RDDM, pool\_size = 75$ |
| TabPFN[Stream] | $context\_window = 1,000, short\_term\_window = 750, long\_term\_window = 250$ |

## K  LLM USAGE

Large Language Models (LLMs) were used to polish the writing of this paper. Technical contributions, experiments, and analyses were carried out by the authors.

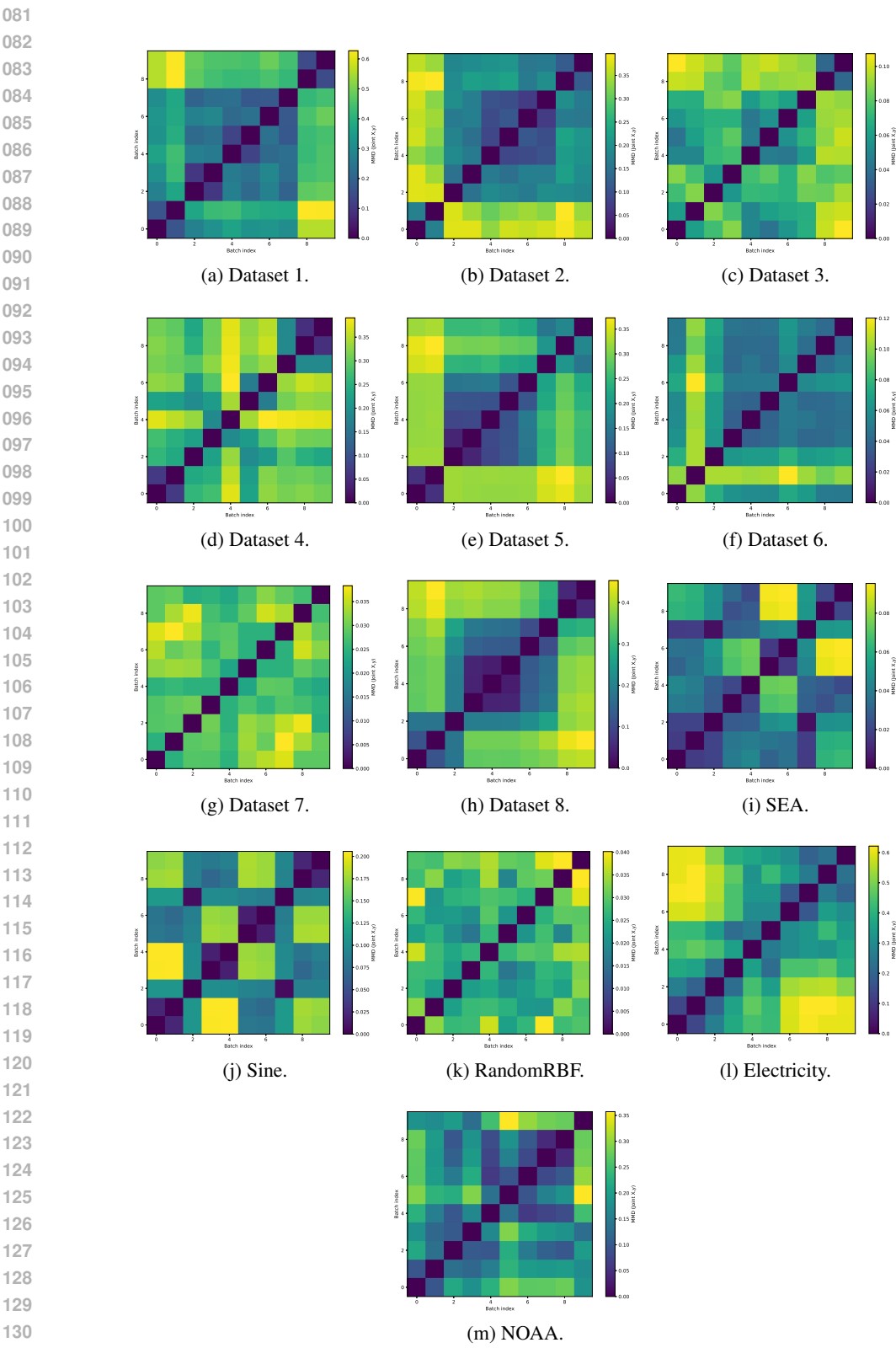

Figure 6: Maximum Mean Discrepancy Heatmaps on datasets.

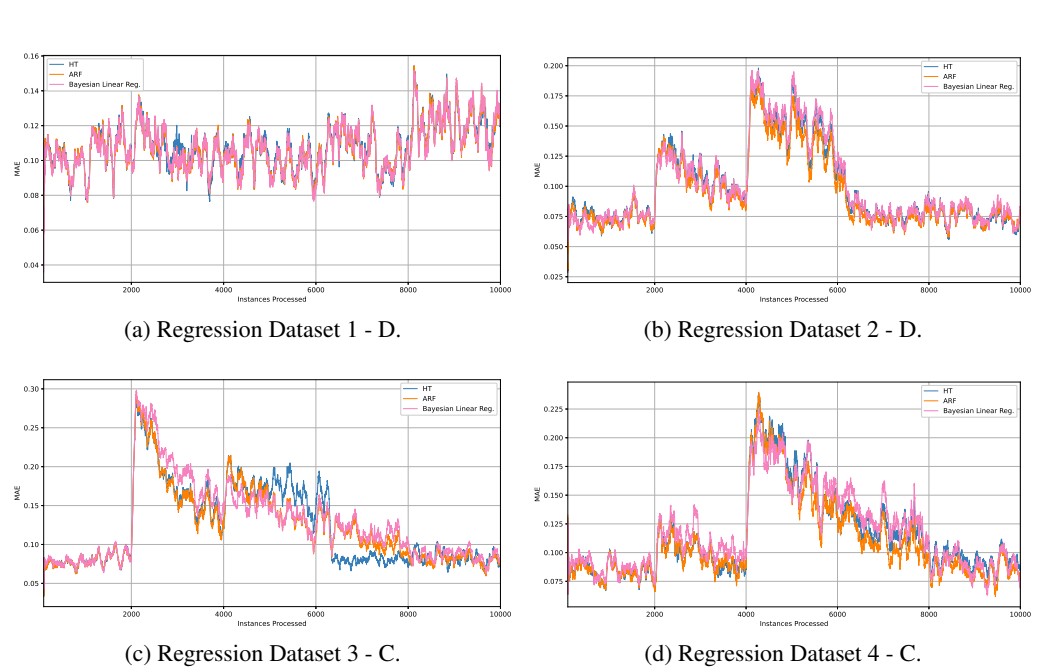

(a) Regression Dataset 1 - D.

(b) Regression Dataset 2 - D.

(c) Regression Dataset 3 - C.

(d) Regression Dataset 4 - C.

Figure 7: Prequential mean absolute error on datasets. Letters refer to the distributional shifts applied to the datasets. D stands for distributional, and C for covariate.

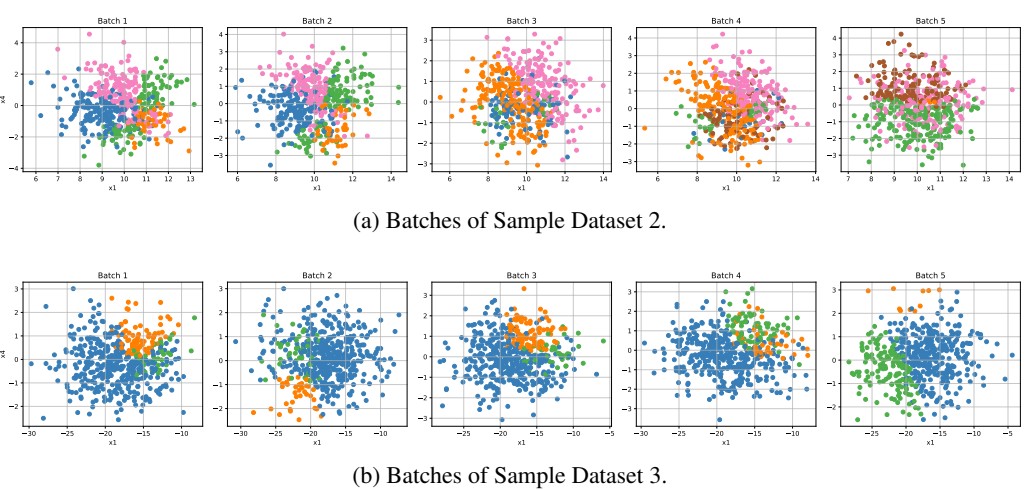

(a) Batches of Sample Dataset 2.

(b) Batches of Sample Dataset 3.

Figure 8: Class distribution across batches for datasets sampled by CaDrift.

