# OpenReview forum: "CaDrift: A Time-dependent Causal Generator of Drifting Data Streams"
_ICLR.cc/2026/Conference — Submitted to ICLR 2026_

### Official Review · Reviewer_v6KE · 2025-10-26

**Soundness:** 3
**Presentation:** 2
**Contribution:** 1
**Rating:** 2
**Confidence:** 3

**Summary:**

This paper introduces CaDrift, a framework for generating synthetic, time-dependent data streams with concept drift, based on Structural Causal Models (SCMs). The goal is to provide a controlled environment for evaluating machine learning models on evolving data. CaDrift simulates various drift types by altering the mapping functions within the SCM and introduces temporal dependencies via Exponentially Weighted Moving Average (EWMA) and autoregressive noise. The authors conduct experiments to evaluate the performance of several streaming classifiers on the generated data and perform stationarity tests to validate the non-iid nature of the output. The paper claims this is the first SCM-based generator to produce time-dependent data with controlled shifts.

**Strengths:**

1. The work addresses a well-recognized need in the data stream community: the lack of versatile and controllable benchmark generators. The ability to simulate specific types of drifts, including those rooted in causal mechanisms, is a valuable goal.

2. The proposed framework is technically sound. The combination of SCMs for structured data generation, classic time-series components (EWMA, AR noise) for temporal correlation, and parameter modulation for drift is a logical and well-implemented approach.

**Weaknesses:**

1. The primary weakness is the limited novelty of the proposed method. CaDrift is essentially an engineering amalgamation of existing, well-known components: SCMs, EWMA, and AR models. While the integration is effective, it does not introduce new machine learning principles or a novel theoretical framework. The contribution feels more like a useful software tool than a fundamental research advancement.

2. The paper heavily emphasizes its "causal" foundation, yet the experimental design fails to demonstrate why this is beneficial. The experiments do not show how the underlying causal structure leads to challenges that are distinct from those in non-causal generators. For instance, there are no experiments showing how models fail on specific types of causal confounding or how a causally-aware model might excel on CaDrift's data. Without such evidence, the "causal" aspect remains an unsubstantiated claim of superiority.

3. The main experimental results (Table 2, Fig. 3) largely confirm a well-established phenomenon in streaming literature: model performance drops after a concept drift, and adaptive models recover faster. The experiments use CaDrift as a testbed but fail to yield any new, deep insights into model behavior. The controllable nature of the generator is underutilized; it could have been used to explore more nuanced questions, such as how the topological location of a drift in the causal graph affects model adaptation.

4. It is not clear that the data generated by CaDrift presents challenges that are fundamentally different from or more realistic than those from widely-used existing generators (e.g., SEA, RandomRBF, Hyperplane). While CaDrift offers more explicit control, the paper does not demonstrate that this control allows for simulating real-world phenomena that other generators cannot. The value proposition over the existing ecosystem of benchmarks is not strongly made.

**Questions:**

See Weaknesses.

---

> ### Author Response · Authors · 2025-11-21
>
> We thank the reviewer for the time to read our work and for the constructive and insightful feedback. Below, we answer the points raised:
> ## W1. Novelty
>
> CaDrift introduces a novel contribution to the data stream literature by enabling controllable structural drift through explicit manipulation of causal mechanisms, something no existing synthetic generator supports. Unlike classical probabilistic generators, which modify only global distributional parameters, CaDrift operates at the level of structural equations, which directly modify the causal mechanism of a node and induce downstream effects throughout its descendants.
>
> By altering structural relationships in the graph, CaDrift can simulate a broad range of drift events, directly grounded to their definition (e.g., $P_t(y \mid X) \neq P_{t+\delta}(y \mid X)$ for distributional shift) in a way that purely statistical generators cannot.
>
> We acknowledge that CaDrift builds upon well-known components (e.g., SCMs, EWMA, AR noise), but the contribution lies in how they are integrated into the first unified framework for causal, time-dependent drift generation. To support this, we added a Maximum Mean Discrepancy (MMD) analysis in Appendix G (Fig. 6), which shows that CaDrift produces non-stationary behavior much closer to real-world datasets, while standard benchmarks remain nearly stationary.
> ## W2. Causal Foundation
>
> The causal foundation of CaDrift enables complex, controlled, and interpretable drift events that cannot be represented by existing iid generators, which provide only a fixed set of predefined shifts. It also provides a richer variety of shift events, as also supported by the new MMD analysis. Thus, the causal foundation enables a broader and more realistic family of drifts that non-causal generators cannot emulate.
>
> We agree that further model-level analysis (e.g., isolating specific confounding patterns or testing causally-aware models) would be valuable, but conducting an exhaustive causal study is beyond the scope of this work -- our evaluation focused on representative stream-learning baselines to demonstrate that CaDrift can effectively generate temporally correlated, drifting data streams rather than to exhaustively benchmark a large variety of models.
>
> ## W3. Model Performance and Insights
>
> The accuracy drops and recoveries observed in Figure 3 confirm that CaDrift induces controllable drift events -- precisely the goal of our work. The intent of this paper is to introduce and validate the framework, not to exhaustively analyze model adaptation. By establishing that CaDrift reproduces expected dynamics, we provide a reliable foundation for future studies to explore more nuanced questions, such as those pointed out by the reviewer (e.g., how drift location and modifications in specific points in the causal graph influence adaptation). Additionally, we did explore the controllable nature of the generator -- we have made some specific modifications to the causal graph of datasets (e.g., modifying target function of nodes, incremental updates on mappers, changing parameters of root mappers, etc.), which have impacted model behavior. Details on how modifications were made to the effect functions of some of the graphs are available in Appendix H, Table 10.
>
> ## W4. Comparison to existing Generators
>
> We understand the reviewer's concern and appreciate the opportunity to argue the advantages of CaDrift over other generators. CaDrift explicitly models causal and temporal mechanisms, where each feature is generated through structural causal relations. This enables interventions on specific nodes or structural functions to produce controlled and interpretable drifts. Such changes correspond to modifying the conditional mechanism $$P(x_j|C_j)$$for a single node $x_j$. Because the system is structural, this perturbation naturally propagates through the graph, altering the marginal distributions $$P(x_k) = \sum_{C_k} P(x_k|C_k)P(C_k)$$for all descendants $k \in \text{descendants}(j)$. This type of localized mechanism shift cannot be simulated by existing synthetic generators, which operate directly on the joint distribution through global perturbations (e.g., centroid shifts, hyperplane rotations, changing thresholds) and do not modify structural mechanisms or induce downstream propagation effects through a causal graph.  Thus, CaDrift can simulate drift phenomena that produce downstream effects in a causal graph, which popular generators cannot achieve.
>
> Moreover, the experiments with limited and delayed supervision available in Appendix E (Table 5) show that, under these scenarios, baseline models exhibit larger drops in accuracy on data streams generated by CaDrift than on existing synthetic benchmarks, which also happens in real-world datasets (Gomes et al.,  2017; Barboza et al., 2025). The MMD analysis reinforces that CaDrift produces temporally evolving, non-stationary distributions closer to those observed in real-world streams.

---

### Official Review · Reviewer_hasu · 2025-10-27

**Soundness:** 3
**Presentation:** 2
**Contribution:** 3
**Rating:** 6
**Confidence:** 4

**Summary:**

The paper proposes a dataset generator for generating causal concept drift datasets that can be used for benchmarking concept drift detection and adaptation methods.

**Strengths:**

- Addresses an important problem, which is well-grounded in the literature. Proper evaluation of concept drift adaptation/detection methods is a long-standing problem in the field. In this context, the paper has the potential to make an impactful contribution.
- Mostly well-written and easy to follow.
- Implementation of the proposed data generator is provided and seems somewhat easy to use. However, I do have a few suggestions on how to further improve it.

**Weaknesses:**

Table 2:
I suggest including some notion of variance. Averages alone are not that meaningful. In particular, when it comes to statistical significance analyses.
Also, I suggest briefly describing the evaluated methods to make the paper more self-contained and more accessible. Right now, only TabPFN is described, and all other methods are only in Table 10 in the appendix.

Section 5.2:
I suggest starting this section by stating the purpose or objective of the evaluation that follows. It becomes clear from reading the section, but right now, the reading flow gets interrupted.

Some suggestions on how to improve the usability of the Python code -- I consider this to be important since the main contribution of the paper is a dataset generator (limited usability will directly affect the impact of the paper):
- Overview of different target functions, as well as more documentation on the usage in the README
- Better documentation (e.g., doc strings) of classes and functions that have to be used by the user. In particular, it would be nice to have type hints for arguments and return values. Not all classes (Mappers, TargetFunctions, CausalGraph) have doc strings!

**Questions:**

See weaknesses

---

> ### Author Response · Authors · 2025-11-21
>
> We want to thank the reviewer for the positive feedback on our work. We appreciate the suggestions and made sure that all of them were implemented in the manuscript and the code. Below, we answer each point.
>
> ## W1. Variance and description of baselines
>
> We have included the standard deviation of classification accuracy in Table 2, and a brief description of each baseline method in Appendix I in the revised manuscript.
>
> ## W2. Section 5.2: State the purpose at the beginning
>
> We have added an opening sentence explaining what is done in Section 5.2: ``In this section, we perform stationarity tests to assess how each component impacts the induction of serial correlation upon samples generated by CaDrift. In Table 3 we show the results of the Ljung-box test...``
>
> ## W3. Enhance Python code and README
>
> We appreciate the reviewer’s suggestions regarding the Python code. We have added comprehensive docstrings to all major classes (including Mappers, TargetFunctions, and CausalGraph) and included type hints for all arguments and return values. We have also expanded the README to provide an overview of the available target functions, clearer usage instructions, and additional examples, including code snippets illustrating how to generate regression datasets and different types of concept drift. These updates substantially improve both usability and clarity of the CaDrift implementation.

---

> > ### Comment · Reviewer_hasu · 2025-11-23
> >
> > Thank you for your rebuttal and the revised paper. The rebuttal and the revision answer my questions and address my concerns. I will raise my score.

---

> > > ### Author Response · Authors · 2025-12-03
> > >
> > > Thank you for the follow-up and for taking the time to re-evaluate our work. We are glad that the rebuttal and revisions addressed your concerns.

---

### Official Review · Reviewer_f36o · 2025-11-10

**Soundness:** 3
**Presentation:** 2
**Contribution:** 2
**Rating:** 4
**Confidence:** 3

**Summary:**

The paper proposes a synthetic stream generator built on time-dependent structural causal models (SCMs). It induces concept drift by drifting SCM mapping functions (distributional/severe shifts) and by adjusting root-node distributions (covariate/local shifts). Experiments show the expected accuracy drop after drift and subsequent recovery across several stream learners.

**Strengths:**

a. This paper claims to be the first drift generator that couples causal structure with explicit temporal dynamics in an SCM framework, going beyond IID or purely probabilistic generators.

b. The proposed method supports distributional, covariate, severe, local drifts with abrupt/gradual/incremental/recurrent patterns and configurable windows.

**Weaknesses:**

a. No quantitative alignment with real drifting datasets or transfer evidence that tuning on CaDrift improves real-world performance.

b. The evaluation only focused on classification accuracy. Little coverage of regression/unsupervised settings, efficiency of generation, or additional evaluation metrics.

**Questions:**

a. Any plan to quantitatively compare CaDrift streams to real datasets (e.g., distribution distances over time, drift detector behavior, or downstream transfer gains)?

---

> ### Author Response · Authors · 2025-11-21
>
> We appreciate the reviewer's time in reading the manuscript and for providing important insights into our work. Below, we address each point and answer the reviewer's question.
>
> ## W1. Quantitative alignment and transfer evidence.
>
> We have performed the ljung-box test on real-world datasets as well in the revised manuscript (Appendix F, Table 7 in the revised manuscript), and confirmed that the behavior of CaDrift is more aligned with real-world than existing synthetic benchmarks: on real-world datasets, all features and target present serial correlation, a behavior replicated by streams generated by CaDrift, but not by popular generators. Furthermore, regarding CaDrift improving real-world performance, that is indeed the natural next step, but tuning CaDrift to downstream tasks is beyond the scope of this work. Instead, we fill a long-standing bottleneck in the data stream literature, which is the lack of synthetic benchmarks that simulate a wide variety of shift events. Many recent works still rely on purely probabilistic synthetic benchmarks with a limited set of predefined shift events (Barboza et al., 2025; Pham et al., 2025; Assis et al., 2025). CaDrift can synthesize data streams with different characteristics, concept difficulties, and drift types, providing a framework to generate diverse synthetic data.
>
> In the revised manuscript, we added a Maximum Mean Discrepancy (MMD) test in Appendix G (more details below), which reinforces that streams generated by CaDrift are more aligned with what is observed in real world. Below we show the highest pairwise MMD across batches achieved, including popular real-world benchmarks:
>
> | Dataset 1 | Dataset 2 | SEA  | Sine | RandomRBF | Electricity | NOAA |
> | :---: | :---:  | :---: | :---:  | :---:  | :---:  | :---:  |
> | 0.63      | 0.40      | 0.10 | 0.21 | 0.04      | 0.62        | 0.36 |
>
> ## W2. Evaluation focused on classification accuracy
>
> We have focused on classification because it is the broader problem covered in data stream and concept drift literature (Lu et al., 2019; Hinder et al., 2024). In addition, the objective of this paper is not to exhaustively evaluate different ML methods, but to present CaDrift and how its drift events affect the performance of classifiers. The accuracy -- especially the prequential accuracy  -- is sufficient to provide these insights and to show the effect of drift events simulated by CaDrift, as when computed alongside a sliding window, allows for the identification of the impact of concept drifts in classifiers' accuracy (Bifet et al., 2015).
> ### Regression, unsupervised settings, efficiency, and additional metrics
>
> The generation of regression datasets is fully supported by CaDrift. We have added a clarifying sentence in the manuscript (l. 240), included examples in the source code for generating regression streams, and added some regression experiments in Appendix H. A thorough evaluation of regression tasks would require additional baselines and metrics beyond those used in this work, and such an analysis would not change the core claims of the paper.
>
> Regarding unsupervised settings, we show that classifier learning is affected when label availability is delayed or limited (Appendix E, Table 5), demonstrating that CaDrift can simulate realistic challenges in partially supervised scenarios. A full evaluation of unsupervised adaptation methods is beyond the scope of this paper.
>
> Finally, concerning efficiency, CaDrift remains computationally practical even though samples are generated sequentially to preserve temporal dependence. The implementation can produce large streams in reasonable time -- e.g., on a ten-dimensional graph, we were able to generate roughly 3,000 samples per second, producing streams of hundreds of thousands of instances within minutes. While the generation process for a single stream is inherently sequential, multiple streams can be generated in parallel.
>
> We have added time benchmarks in the source code to document this.
>
> ## Q: Quantitative comparison between CaDrift streams to real datasets
>
> Thank you for the suggestion. As mentioned above, we have added a quantitative comparison between CaDrift and real datasets using MMD (Appendix G). These experiments show that CaDrift exhibits temporally varying distributional shifts that closely mirror the non-stationarity observed in real streams, whereas standard synthetic generators remain nearly stationary. Assessing drift-detector behavior and downstream transfer gains are beyond the scope of the present paper, which focuses on introducing CaDrift and validating that it produces realistic drifting data.

---

### Author Response · Authors · 2025-12-03

We would like to thank the reviewers for their time, comments, and feedback on our work. Unfortunately, due to the recent incident, most of the reviewers did not have the opportunity to follow up on the initial reviews. Still, we want to express our gratitude for the suggestions, which were valuable in helping us strengthen the paper.

We also thank the area chair for the time and care dedicated to evaluating our submission under the current circumstances. During the rebuttal stage, we expanded and clarified key aspects of the work. Specifically, we added a more detailed distributional analysis using MMD to better characterize the impact of generated shift events, and we included regression experiments demonstrating that the generator remains challenging across both classification and regression settings. These additions reinforce CaDrift's generality and practical value in addressing a long-standing bottleneck in data streams: the lack of challenging datasets with known and controllable shift events.

We hope these clarifications and revisions made during the rebuttal stage better frame the contribution and potential impact of our work, and we are grateful for your consideration in the final decision.

---

### Meta-Review · Area_Chair_Rmjm · 2026-01-08

**Summary:**

There is general review agreement that the paper addresses an important gap in data stream research with technically sound and practically useful framework. Core concerns involve the perceived novelty, with some reviewers viewing CaDrift as an engineering integration of known components rather than a theoretical advance, and questions about whether the causal foundation materially improves realism or experimental insights over existing generators. I think the authors' rebuttal has addressed some of these points. Overall, the rebuttal has mitigated review concerns. Reviewer 2 explicitly updated their rating after rebuttal.

Given the mixed ratings (4, 6, 2), I recommend a borderline reject, as its theoretical novelty is limited.

**Reviewer Concerns:**

Reviewer 1 review is relative short, and most concerns are clarified after rebuttal.

Reviewer 2 acknowledged the asked concerns are fixes and has raised rating to 6.

Reviewer 3 noted the work more of an engineering integration than a fundamental research advance, and no experiments directly show how causal structure produces unique challenges for learning models.

**Reviewer Scores:**

Reviewer 1 likely would remain original rating.

Reviewer 2 already indicated the raised rating to 6 after rebuttal.

Reviewer 3 would probably would remain original rating due to lingering concerns about theoretical contribution.

---

### Decision · Program_Chairs · 2026-01-26

Reject